## [Transparent Peer Review file · Nature Communications]

Unlocking Large-Area Free-Standing MOF-Glasses for Molecular Sieving Gas Separation Membranes

Corresponding Author: Dr Alexander Knebel

Version 0:

Reviewer comments:

Reviewer #1

(Remarks to the Author)

This manuscript presents a highly significant and meticulously executed study on the fabrication of large-area, free-standing MOF-glass membranes for gas separation. The work successfully addresses a major bottleneck in the field—the inherent brittleness and cracking of MOF-glasses—through a clever combination of empirical substrate selection (matching CTE), and an optimized annealing protocol. The results are outstanding: the demonstrated perfect molecular sieving of CH₄, with undetectable permeability, is a remarkable achievement that underscores the defect-free, grain-boundary-free nature of the prepared agZIF-62 glass films. The methodology is well-described, the data (XRD, SEM-EDX, permeation tests) are convincing and fully support the claims, and the discussion is thoughtful. This work represents a substantial step forward towards the practical application of MOF-glasses in high-precision separation membranes. It is recommended for publication after minor revisions addressing the points below.

1. As shown in Fig. 2b, the crystalline structure completely disappears after the transformation of ZIF-62 crystalline seeds into a ZIF-62 glassy membrane. What is the underlying reason for this phenomenon?
2. How does the pore size distribution change after the conversion of ZIF-62 crystalline seeds into a ZIF-62 glassy membrane? The pore size distribution of MOFs can be further determined.
3. Membrane thickness is a critical parameter for gas separation applications. Can the thickness of the MOF-glass membranes be controlled using your fabrication method? What is the relevant influence?

Reviewer #2

(Remarks to the Author)

In this manuscript, Knebel et al., reported the design and engineering of a MOF glass membrane for gas separation. By carefully optimizing each step of the process – from the initial substrate choice to the materials melting and final performance validation, a dense and free-standing ZIF-62 glass membrane was successfully fabricated. The membrane microstructure was investigated in details. Benefiting from the absence of grain boundaries, an exceptional selectivity for CH₄ and smaller gas specie was achieved. The concept of fabricating free-standing gas separation membranes from MOF glass presents a promising approach, with the potential to inspire new strategies for creating high-performance molecular sieving membranes. I would like to recommend its publication in Nature Communication.

There are some minor comments to be considered as following:

1. MOF glass based molecular sieving membranes have become a research hotspot in the past decade, what are the principal advantages of MOF glass membranes over polycrystalline MOF membranes? A detailed comparison is suggested to be provided.
2. This manuscript describes the fabrication of a “large-area” MOF glass membrane with a diameter of 18 mm. It would be helpful if the authors could clarify the benchmark for this description, as this scale is common for polycrystalline MOF membranes. What is “large-area” relative to?
3. In line 243, Page 9, the authors claimed that “our experimental value of the real selectivity $\alpha_{real}(N_2/CO_2) = 22.....$ ”. We observed that the CO₂ permeability of the resulting glass membrane was higher than that of N₂ (Fig. 4b), thus the $\alpha_{real}(N_2/CO_2)$ should be $\alpha_{real}(CO_2/N_2)$. Please carefully check it.
4. The mechanical stability of membrane materials is important for real-world application, what about the stability of ZIF-62 glass membrane?
5. What is the pore channels size of the resulting ZIF-62 glass membrane?

Reviewer #3

(Remarks to the Author)

This paper presents a significant advancement in the field of MOF-based membranes for gas separation by addressing key limitations in the production and application of MOF glasses, specifically the issues of cracking during processing and the difficulty in creating large, free-standing membranes. I recommend acceptance of this article for publication, as the research addresses a significant challenge in the field and the results demonstrate a promising solution for producing high-performance MOF-glass membranes for gas separation. The clarity of presentation and thoroughness of the characterization further support the value of this work. However, I urge the authors to temper their enthusiasm and acknowledge the limitations of their findings. The study convincingly demonstrates the feasibility of their approach for ZIF-62 glass membranes, but also not more as other ZIFs, MOFs, or related materials are not included. The authors should frame the study as a proof-of-concept and carefully revise the manuscript to remove any overly generalized statements regarding the broader applicability of their study.

Reviewer #4

(Remarks to the Author)

The authors present an innovative study on free-standing glassy MOF membranes for molecular sieving based on ZIF-62. They were able to produce a crack-free and bubble-free membrane with accessible pore structure of approximately 330 microns in thickness. The authors circumvented many of the problems associated with melt-quenched approaches by directly annealing the solidified melt. In addition, the authors placed their glassy MOF membrane within an appropriate ring structure, creating a robust, free-standing membrane with a large surface area that is suitable for permeation tests in a Wicke-Kallenbach setup with online gas chromatography coupling. Their approach greatly simplifies processing and has huge potential for technology transfer. Gas permeation was measured for single gases (He, H₂, CO₂, N₂ and CH₄) and binary gas mixtures (H₂/CO₂, CO₂/N₂, He/CO₂, H₂/CH₄) with their kinetic diameters in the range of 2.5 to 4.0 Angstrom. Although, the overall gas permeability is low, due to the relatively thick membrane, the selectivities are comparable to literature, and in the case of H₂/CH₄ separation they exceed literature values clearly in an impressive manner. The H₂/CH₄ selectivity is infinity. If methane (CH₄) is permeating at all, it is below the detection limit of the authors' state-of-the-art setup. Additionally, the authors present ideas for the design of membrane modules, including flexible forms. In summary, the authors set a new benchmark for the H₂/CH₄ selectivity with glassy MOF membranes, bringing technological applications of this membrane class closer. The results hold considerable potential significance, promising to advance understanding in the field and contribute meaningfully to its progression. The conclusions are well supported. The manuscript would, however, benefit from some minor revisions, including the following:

1. The manuscript would benefit greatly from a more explicit comparison of the authors' results with those of the current state of the art, which has developed since Wang et al.'s early paper on the subject, A MOF Glass Membrane for Gas Separation, *Angew. Chem. Int. Ed.* 59 (2020), 4365-4369, <https://doi.org/10.1002/anie.201915807>. These authors used melt-quenched ZIF-62 and achieved a similar CO₂/N₂ selectivity at one order of magnitude higher permeability at an overall thickness of the membrane of 20 to 30 microns compared to a thickness of 330 microns in the authors' manuscript.
2. "Large area" should be quantified and compared to the current state of the art.
3. Permeation and selectivities should be quantified and compared to the current state of the art.
3. Is there a rational behind the selection and order of permeability test displayed in Figure 4c,d?
4. Why the possible binary gas mixtures were restricted to four (i.e. H₂/CO₂, CO₂/N₂, He/CO₂, H₂/CH₄)? Including He/CH₄ and CO₂/CH₄ separation could enhance the impact of the manuscript.
5. Line 310ff: In the context of the mentioned "perovskite-based membranes", it would be helpful to direct the reader by citing two relevant reviews.
6. Supporting material, section 1.7: Avoid duplicating content in the second paragraph and use a consistent acronym for X-ray spectroscopy (e.g. EDS or EDX).

Reviewer #5

(Remarks to the Author)

This work reports a type of ZIF-62 glass membranes, fabricated through thermal expansion coefficient matching between the MOF glass and the processing devices. The authors observed impressive selectivity particularly for bulkier gas molecules, and also discussed the possible pathway forward towards the practical applications. Overall, this contribution addresses some of the critical challenges associated with the MOF glass, particularly ZIF-62, and the fabricated membrane could offer some solutions to challenging separations. The paper can be considered for publication in *Nature Communications*, after some further improvement.

1. The authors claim the preservation of mechanically robust, cracking free membrane is originated from the CTE matching between MOF glass and Al foil. This could be the reason but more investigation will be required to fully support this claim. For example, does the ZIF-62 attach strongly with the foil after sintering? Or the mechanism could be just a low energy interface that does not provide additional stress to MOFs? More types of foils/substrates with different CTE would be welcomed.

2. It is impressive to see the high selectivity of this membrane, which addresses one of the critical challenges in separation. There are some recent arguments on the use of W-K setup for membrane permeation test - for example, the sweep gas of He can back diffuse to the feed side, and this will lead to a higher measured membrane permeability than the actual value. Could the author provide further discussion on this?

3. It would be interesting to see if the membrane performance variance with different thickness, and/or different benzimidazolate ratio.

4. There are multiple typos in the manuscript, better to revise them.

Version 1:

Reviewer comments:

Reviewer #1

(Remarks to the Author)

The revision and the response from the authors have addressed the questions from the referee. It can be recommended for publication in this current version.

Reviewer #2

(Remarks to the Author)

The authors have well addressed the reviewers' comments. I think the revised manuscript can be accepted by Nature Communn.

Reviewer #3

(Remarks to the Author)

This is an excellent paper that provides a practical and scientifically rigorous solution to a long-standing problem in MOF-glass membrane fabrication. The authors successfully demonstrate the production of large-area, crack-free, self-supported MOF-glass membranes with outstanding molecular sieving capabilities. The methodology is well-detailed, and the results are compelling. Especially I like the detailed analysis and clear presentation in this paper. I have only the two comments for improving this strong paper:

The authors states that methane permeability was so low that it was "undetectable" by their gas chromatography setup. Therefore, please include a quantitative upper bound for the methane permeability, rather than simply stating it's "undetectable".

The gas permeation experiments were conducted at room temperature (22°C) and 1 atm. A brief discussion or ideally, some data on how the separation performance (permeability and selectivity) changes with variations in temperature and pressure would provide a more complete (impactful) picture of the membrane's versatility and applicability in different industrial scenarios.

Reviewer #4

(Remarks to the Author)

The authors addressed my concerns in a reasonable manner. The manuscript is recommended for publication.

Reviewer #5

(Remarks to the Author)

The authors have addressed the comments raised by the last round of review - now the paper can be accepted

Rebuttal Letter

We would like to thank all five referees, for the time they dedicated to reviewing our manuscript and for their general evaluation of our work. The referees' comments – to which our point-by-point responses are provided below – helped us strengthen the manuscript. The referees' comments are formatted in *italic*, our responses in **green**, changes in the manuscript in **yellow**, and parts removed from the main text – in ~~strikethrough red~~.

The key revisions include:

- Figure 1c was transformed from a scheme into a graph.
- We have tested two additional inert substrates for ZIF-62 melting – borosilicate float glass BF33 and pure gold. As expected from the CTE mismatch, both substrates resulted in significant glass cracking, confirming that the initially suggested aluminum remains the optimal choice. The results and corresponding discussion have been incorporated into the manuscript.
- We have deepened the discussion on substrate- a_2 ZIF-62 interactions and supported the attribution of the observed cracking behavior primarily to residual thermal stresses through quantitative estimations.
- More context and detail have been added to the introduction, including a thorough discussion of the potential benefits of glassy MOF membranes over their crystalline counterparts. The remainder of the introduction has been slightly restructured and shortened to improve flow and clarity and to avoid repetition.
- Permeation data for He/CH₄ was added and small but significant typos (for example N₂/CO₂ instead of CO₂/N₂) were corrected.
- Data for He/CH₄ was added to the new Figure 4c and 4d with the expected results, no CH₄ was passing through the membrane.
- The manuscript now includes a detailed qualitative summary of the reported MOF-glass membranes from literature and their tested results for various gas separations, which have been collected in a table, plotted and discussed accordingly.
- The material and methods moved from the SI to the main manuscript to fit the style of Nat. Commun.
- Other minor revisions are covered in our point-by-point responses.

We believe the revised manuscript has improved a lot and we thank everyone involved again for their constructive feedback.

On behalf of all authors,

Dr. Alexander Knebel

Reviewer #1:

This manuscript presents a highly significant and meticulously executed study on the fabrication of large-area, free-standing MOF-glass membranes for gas separation. The work successfully addresses a major bottleneck in the field—the inherent brittleness and cracking of MOF-glasses—through a clever combination of empirical substrate selection (matching CTE), and an optimized annealing protocol. The results are outstanding: the demonstrated perfect molecular sieving of CH₄, with undetectable permeability, is a remarkable achievement that underscores the defect-free, grain-boundary-free nature of the prepared agZIF-62 glass films. The methodology is well-described, the data (XRD, SEM-EDX, permeation tests) are convincing and fully support the claims, and the discussion is thoughtful. This work represents a substantial step forward towards the practical application of MOF-glasses in high-precision separation membranes. It is recommended for publication after minor revisions addressing the points below.

We are sincerely grateful to the referee for their positive and highly encouraging evaluation of our manuscript. Below, we provide a point-by-point response to all comments.

1. As shown in Fig. 2b, the crystalline structure completely disappears after the transformation of ZIF-62 crystalline seeds into a ZIF-62 glassy membrane. What is the underlying reason for this phenomenon?

X-ray diffraction (XRD) relies on constructive interference from regularly spaced lattice planes. However, when a crystalline solid – or a mixture of solids thereof – is melted into a glass, the long-range periodic atomic order is lost. Indeed, the quenching of a glass results in the kinetically driven freezing of a long-range disordered melt structure, thereby preventing the atoms from re-ordering into thermodynamically more stable crystalline structures. Thus, the absence of periodicity eliminates sharp Bragg reflections, leaving only broad halos in place of distinct XRD peaks. This behavior is typical for all glasses and particularly for melt-quenched ZIF-62, in agreement with all relevant previous reports (e.g., [10.1126/sciadv.aao6827](https://doi.org/10.1126/sciadv.aao6827), [10.1038/s41563-023-01738-3](https://doi.org/10.1038/s41563-023-01738-3), [10.1063/5.0031941](https://doi.org/10.1063/5.0031941), [10.1016/j.jnoncrysol.2019.119806](https://doi.org/10.1016/j.jnoncrysol.2019.119806), [10.1038/s41467-024-49428-1](https://doi.org/10.1038/s41467-024-49428-1)).

2. How does the pore size distribution change after the conversion of ZIF-62 crystalline seeds into a ZIF-62 glassy membrane? The pore size distribution of MOFs can be further determined.

We thank the referee for this comment. Since ZIF-62 is a well-studied glass-former among MOFs, its pore size distribution and its evolution have been investigated in detail in previous works, e.g. physisorption experiments by Frenzel-Beyme et al. in [10.1038/s41467-022-35372-5](https://doi.org/10.1038/s41467-022-35372-5) or PALS data reported by Wang et al. [/10.1002/anie.201915807](https://doi.org/10.1002/anie.201915807). Nevertheless, we would like to respectfully note that while the pore size is an important characteristic of a porous material, parameters like channel size or pore window size provide more comprehensive information for the diffusive gas separation, as performed in our membrane study, since, for instance, a large pore's guest accommodating ability might be severely limited by a narrow pore window. It is equally applicable to the disordered glasses, where preservation of the accessible channels plays critical role – even more important than preserving the large pores – since they determine the molecular pathways and therefore the permeation ability.

Thus, the pore channel size distribution in ZIF-62 glasses prepared by following the same initial synthetic and melting protocols have been investigated in detail via HR-TEM in our previous study in *Nat. Mater.* ([10.1038/s41563-023-01738-3](https://doi.org/10.1038/s41563-023-01738-3)) and demonstrated that our melting methodology allows for preserving the channels accessible for the guest molecules upon glass formation. Overall, due to the non-ordered glassy nature, pore

channel sizes varying from approximately 1,7 Å to 3,4 Å with an average pore-limiting diameter of $2,7 \pm 0,5$ Å for a melt-pressed sample (as we used here) have been observed in the normal distribution for the analogous to the current study samples. This distribution explains the gradual loss of permeability in the row He ($d_{kin} = 2.6$ Å), H₂ ($d_{kin} = 2.89$ Å), CO₂ ($d_{kin} = 3.3$ Å), N₂ ($d_{kin} = 3.64$ Å), and CH₄ ($d_{kin} = 3.8$ Å). We have now included this data and discussed it accordingly:

[...] This result is supported by our previous preliminary work, where an absolute exclusion of ethane ($d_{kin}(\text{C}_2\text{H}_6) = 4.4$ Å) diffusion was achieved after liquid-state processing of a_gZIF-62, suggesting that this is likely possible, also for CH₄ and other relatively large gas molecules. In particular, the pore channel size distribution in such, but millimeter-scale glasses prepared by following the same initial synthetic and melting protocols have been investigated in detail via HR-TEM: pore channel sizes varying from approximately 1.7 Å to 4 Å with an average pore-limiting diameter of 2.7 Å have been observed in the normal distribution for the analogous to the current work samples.^[11] The above-demonstrated permeation results are indeed in agreement with this data: a disordered glassy nature results in a relatively broad pore channel size distribution but the ≥ 3.8 Å (potentially allowing for CH₄ ($d_{kin} = 3.8$ Å) permeation) fraction is almost negligible compared to the smaller sizes. At the same time, the 3.64 Å - 3.8 Å fraction, enabling selective N₂ permeation, is more present and therefore notable N₂ signal is observed. [...]

3. Membrane thickness is a critical parameter for gas separation applications. Can the thickness of the MOF-glass membranes be controlled using your fabrication method? What is the relevant influence?

We thank the referee for this comment. Our method indeed allows for the thickness control of the resulting membranes via two ways – direct and indirect. The direct way involves initial control of the amount of the crystalline powder used for the glass formation per substrate area – the more powder is used, the thicker membrane it will result in. The indirect way, on the other hand, involves post-processing techniques such as polishing to reduce the thickness – as has been shown before in (10.1038/s41467-024-49428-1) ZIF-62 glass is prone to the polishing techniques standard for glass handling. Those methods are straightforward and do not rely on the complicated membrane thickness control techniques demonstrated for the supported MOF glass membranes, e.g., precise but time- and energy-demanding cathodic deposition-assisted method in 10.1002/anie.202401817. We have updated the relevant discussion as follows:

As we demonstrated before, the material easily withstands polishing and other types of conventional glass treatment, such as liquid-state shaping, so thinner layers can be indeed fabricated by such processing techniques. Moreover, the initial membrane thickness can be modified within a certain range by controlling the amount of the crystalline powder per substrate area before melting.¹¹⁻¹⁴

Then, as demonstrated in other studies on MOF glass gas separation membranes (e.g., 10.1038/s41467-025-56295-x), reducing the thickness leads to some increase in permeance while sacrificing selectivity. This trade off is intuitive and is anticipated for our membrane as well; yet, the need to maintain the mechanical integrity of the free-standing glass membrane would not allow for the significant decrease in the membrane's thickness (e.g., like up to 10 times in the abovementioned 10.1002/anie.202401817). While significant changes would indeed be expected when decreasing the membrane thickness down to at least the tens-of-micrometers range, such thicknesses cannot yet be achieved for neat free-standing MOF glass membranes by either our or other existing methods and are therefore outside the scope of this study.

Reviewer #2:

In this manuscript, Knebel et.al., reported the design and engineering of a MOF glass membrane for gas separation. By carefully optimizing each step of the process – from the initial substrate choice to the materials melting and final performance validation, a dense and free-standing ZIF-62 glass membrane was successfully fabricated. The membrane microstructure was investigated in details. Benefiting from the absence of grain boundaries, an exceptional selectivity for CH₄ and smaller gas specie was achieved. The concept of fabricating free-standing gas separation membranes from MOF glass presents a promising approach, with the potential to inspire new strategies for creating high-performance molecular sieving membranes. I would like to recommend its publication in Nature Communication.

There are some minor comments to be considered as following:

The authors thank the referee for the positive evaluation of our work. Below, we provide a point-by-point response to all comments.

1. MOF glass based molecular sieving membranes have become a research hotspot in the past decade, what are the principal advantages of MOF glass membranes over polycrystalline MOF membranes? A detailed comparison is suggested to be provided.

We thank the referee for their suggestion. The principal advantages of MOF glass membranes over polycrystalline MOF membranes are now summarized in the introduction as follows:

Recently, melt-quenched amorphous framework glasses have become a special area of interest for the separation processes with several potential benefits over their crystalline counterparts. First, framework glasses can be shaped and processed from powders in their liquid state to form a free-standing layer, and therefore do not require complicated thin film growing techniques on expensive ceramic substrates, commonly used for MOFs and COFs. Second, the universal processability of glasses allows for liquid-state shaping and further polishing, cutting, etc., to achieve the desired size and shape. Further, most importantly for membranes, the ability of glass to form a monolithic layer hinders grain boundary diffusion and therefore higher selectivity can be achieved.^{2,5} Finally, unique for MOF-derived glasses, their porosity can be tuned by varying the processing parameters to obtain the targeted size of the molecular-sieving windows.

The rest of the introduction has been slightly restructured and shortened accordingly for the flow and clarity and also to avoid repetitions.

2. This manuscript describes the fabrication of a “large-area” MOF glass membrane with a diameter of 18 mm. It would be helpful if the authors could clarify the benchmark for this description, as this scale is common for polycrystalline MOF membranes. What is “large-area” relative to?

We thank the referee for this comment and agree that this description needs clarification. Indeed, more mature polycrystalline MOF membranes can be commonly fabricated within this and sometimes larger size range. In this work, the “large-area” remark refers, first of all, to the free-standing neat MOF glass layers which have been complicated to fabricate even at the demonstrated here centimeter-scale. Second, this remark further refers to the scaling possibilities that we envision for this type of membranes particularly – e.g., modular designs in Figure 6. We have now clarified it throughout the text, for instance in the following sections:

Despite the great promise, obtaining neat and free-standing MOF glass membranes of a larger area remains challenging. [...] As a result, achieving at least typical membrane area of the polycrystalline MOF membranes – which is by itself commonly limited to the centimeter scale – while preserving porosity and therefore performance characteristics in MOF glasses remains highly problematic. Moreover, in the absence of such established procedures further scaling considerations – potentially unlocking industrially-relevant meter-scale membrane areas – remain unexplored.

[...]

Specifically, first we were able to produce large-area centimetre-scale, free-standing MOF glass membrane [...] As an outlook we will provide strategies for the further development of MOF-glass membranes towards real-world application, including a perspective approach for their upscaling to the industrially relevant membrane areas.

3. In line 243, Page 9, the authors claimed that “our experimental value of the real selectivity $\text{areal}(N_2/CO_2) = 22\dots\dots$ ”. We observed that the CO_2 permeability of the resulting glass membrane was higher than that of N_2 (Fig. 4b), thus the $\text{areal}(N_2/CO_2)$ should be $\text{areal}(CO_2/N_2)$. Please carefully check it.

We thank the reviewer for pointing this out. We did indeed refer to a CO_2/N_2 separation throughout the manuscript. Writing N_2/CO_2 was a typo, but also a significant error. We corrected it everywhere.

4. The mechanical stability of membrane materials is important for real-world application, what about the stability of ZIF-62 glass membrane?

The mechanical stability of the membrane material is indeed of crucial importance for concrete applications. The mechanical properties of hybrid glasses compare less favorably with respect to conventional inorganic glass families, for which elastic moduli, hardness, and fracture toughness are about an order of magnitude larger (compare 10.1038/s41467-020-16382-7 and 10.1111/j.1551-2916.2007.01945.x). Nevertheless, they prove sufficient for the needs of gas separation applications; this does not prevent their processing (polishing, handling; see our work 10.1038/s41467-024-49428-1) nor their shaping and loading resistance to achieve larger-scale membranes, as presented in this study and highlighted throughout the text. Moreover, this task is notably made easier thanks to the sufficiently large thicknesses obtained (330 μm), which associate high gas separation selectivity with preserved material integrity during use.

However, further scaling of such thin layers as is would inevitably and undoubtedly lead to breakage due to their non-flexible nature. For that reason, we would like to again stress out the point that further scaling considerations must always acknowledge this fact and suggest solutions relevant to this particular system – as proposed, for instance, with the modular design in Figure 6, which could simultaneously (1) ensure mechanical robustness of the overall membrane, (2) make it easily transportable if made with flexible (e.g. resin) connections, and (3) allow to change the modules independently if needed and therefore prolonging the life cycle of the whole system.

5. What is the pore channels size of the resulting ZIF-62 glass membrane?

We thank the referee for this comment – indeed, this important information was missing in the report. We thank the referee for this comment. The pore channel size distribution in ZIF-62 glasses prepared by following the same

initial synthetic and melting protocols have been investigated in detail via HR-TEM in our previous study in Nat. Mater. (10.1038/s41563-023-01738-3) and demonstrated that our melting methodology allows for preserving the channels accessible for the guest molecules upon glass formation. Overall, due to the non-ordered glassy nature, pore channel sizes varying from approximately 1,7 Å to 3,4 Å with an average pore-limiting diameter of $2,7 \pm 0,5$ Å for a melt-pressed sample (as we used here) have been observed in the normal distribution for the analogous to the current study samples. This distribution explains the gradual loss of permeability in the row He ($d_{\text{kin}} = 2.6$ Å), H₂ ($d_{\text{kin}} = 2.89$ Å), CO₂ ($d_{\text{kin}} = 3.3$ Å), N₂ ($d_{\text{kin}} = 3.64$ Å), and CH₄ ($d_{\text{kin}} = 3.8$ Å). We have now included this data and discussed it accordingly:

[...] This result is supported by our previous preliminary work, where an absolute exclusion of ethane ($d_{\text{kin}}(\text{C}_2\text{H}_6) = 4.4$ Å) diffusion was achieved after liquid-state processing of a_gZIF-62, suggesting that this is likely possible, also for CH₄ and other relatively large gas molecules. In particular, the pore channel size distribution in such, but millimeter-scale glasses prepared by following the same initial synthetic and melting protocols have been investigated in detail via HR-TEM: pore channel sizes varying from approximately 1.7 Å to 4 Å with an average pore-limiting diameter of 2.7 Å have been observed in the normal distribution for the analogous to the current work samples.^[11] The above-demonstrated permeation results are indeed in agreement with this data: a disordered glassy nature results in a relatively broad pore channel size distribution but the ≥ 3.8 Å (potentially allowing for CH₄ ($d_{\text{kin}} = 3.8$ Å) permeation) fraction is almost negligible compared to the smaller sizes. At the same time, the 3.64 Å - 3.8 Å fraction, enabling selective N₂ permeation, is more present and therefore notable N₂ signal is observed. [...]

Reviewer #3:

This paper presents a significant advancement in the field of MOF-based membranes for gas separation by addressing key limitations in the production and application of MOF glasses, specifically the issues of cracking during processing and the difficulty in creating large, free-standing membranes. I recommend acceptance of this article for publication, as the research addresses a significant challenge in the field and the results demonstrate a promising solution for producing high-performance MOF-glass membranes for gas separation. The clarity of presentation and thoroughness of the characterization further support the value of this work.

We would like to thank the referee for the positive evaluation of our work.

However, I urge the authors to temper their enthusiasm and acknowledge the limitations of their findings. The study convincingly demonstrates the feasibility of their approach for ZIF-62 glass membranes, but also not more as other ZIFs, MOFs, or related materials are not included. The authors should frame the study as a proof-of-concept and carefully revise the manuscript to remove any overly generalized statements regarding the broader applicability of their study.

We thank the referee for this comment and agree that we might have included some overstatements throughout the text. We have carefully revised the text accordingly and pasted some examples of our corrections below. Yet we would like to note that although this study is indeed limited to ZIF-62 as the most straightforward example – which is now stated more clearly – we remain hopeful that this work will inspire the researchers from the field to employ this or other systematic fabrication approaches, carefully considering both reticular and glassy sides of such systems, when also looking into other glass-forming MOF systems. On the other hand, we fully understand that MOF-glass membranes (including ZIF-62) remain at the premature stage due to the overall field's infancy, frequently conflicting published data, and complications related to the system itself. To this end, we transparently outline the remaining challenges and uncertainties within the outlook section and – to the best of our experience – suggest the ways of how those can be approached.

Relevant revisions are exemplified in – but not limited to – the following parts:

Abstract:

[...] Here, we **present suggest** a solution to overcome these limitations and demonstrate that - by selecting suitable materials particularly fitting thermal and mechanical behavior to those of MOF-glass – it is possible to process large, crack-free MOF-glass sheets, **as demonstrated on the example of the well-known MOF-glass former ZIF-62.** [...] We conclude this work by giving a brief outlook of the remaining challenges and perspectives for MOF glasses in a view of gas separation membranes, **envisioning transferability of our approach to other glass-forming systems and their scaling perspectives.**

Introduction:

[...] In this work, we demonstrate a method to achieve larger free-standing MOF glasses from crystalline MOFs (c.f. **Figure 1**) **on the example of meltable zeolitic imidazolate framework ZIF-62.** [...]

[...] As an outlook **we suggest that this or similar – but always systematic, taking both reticular and glassy nature of such systems into account – approaches can be transferred to other glass formers within MOF family.** [...]

Conclusions and outlook:

[...] ~~As demonstrated above,~~ MOF glass membranes possess tremendous potential for high precision separation technologies as they exhibit very sharp molecular-sieving cut-offs. ~~However, handling these materials is still a big challenge, and the herein presented approach offers a solution to this challenge.~~ The high selectivity is achievable because ~~we~~ **the system** combines the well-defined porosity in MOFs with the processability of glasses, allowing for the fabrication of monolithic defect-free layers without grain boundaries. [...] **We envision that systematic fabrication and processing considerations within this work can be applied to other MOF-glass-formers aside from ZIF-62 – in such cases, intrinsic properties and behaviors of these specific systems must be always taken into account.** However, such complex materials at the early stages inevitably raise some challenges. [...]

Reviewer #4:

The authors present an innovative study on free-standing glassy MOF membranes for molecular sieving based on ZIF-62. They were able to produce a crack-free and bubble-free membrane with accessible pore structure of approximately 330 microns in thickness. The authors circumvented many of the problems associated with melt-quenched approaches by directly annealing the solidified melt. In addition, the authors placed their glassy MOF membrane within an appropriate ring structure, creating a robust, free-standing membrane with a large surface area that is suitable for permeation tests in a Wicke-Kallenbach setup with online gas chromatography coupling. Their approach greatly simplifies processing and has huge potential for technology transfer. Gas permeation was measured for single gases (He, H₂, CO₂, N₂ and CH₄) and binary gas mixtures (H₂/CO₂, CO₂/N₂, He/CO₂, H₂/CH₄) with their kinetic diameters in the range of 2.5 to 4.0 Angstrom. Although, the overall gas permeability is low, due to the relatively thick membrane, the selectivities are comparable to literature, and in the case of H₂/CH₄ separation they exceed literature values clearly in an impressive manner. The H₂/CH₄ selectivity is infinity. If methane (CH₄) is permeating at all, it is below the detection limit of the authors' state-of-the-art setup. Additionally, the authors present ideas for the design of membrane modules, including flexible forms. In summary, the authors set a new benchmark for the H₂/CH₄ selectivity with glassy MOF membranes, bringing technological applications of this membrane class closer. The results hold considerable potential significance, promising to advance understanding in the field and contribute meaningfully to its progression. The conclusions are well supported. The manuscript would, however, benefit from some minor revisions, including the following:

We thank the referee for their positive and encouraging evaluation of our manuscript. Below, we provide a point-by-point response to all comments.

1. *The manuscript would benefit greatly from a more explicit comparison of the authors' results with those of the current state of the art, which has developed since Wang et al.'s early paper on the subject, A MOF Glass Membrane for Gas Separation, Angew. Chem. Int. Ed. 59 (2020), 4365-4369, <https://doi.org/10.1002/anie.201915807>. These authors used melt-quenched ZIF-62 and achieved a similar CO₂/N₂ selectivity at one order of magnitude higher permeability at an overall thickness of the membrane of 20 to 30 microns compared to a thickness of 330 microns in the authors' manuscript.*

We thank the referee for this comment. Our manuscript now includes a detailed summary of the state-of-the-art MOF-glass gas separation membranes (Table S3 and Figure S5). Although quantitatively the “infinite” methane selectivity demonstrated in our work is outstanding within the field, we believe that, at the current stage of development, qualitative overview and not the quantitative benchmarking is more informative for the MOF-glass-membranes field. Our reasoning behind such considerations is now included in the main text as follows:

[...] Straightforward comparison of this data with other MOF-glass membranes is complicated by the fact that – as demonstrated in the introduction – currently reported membranes vary significantly in their type (pristine or composited with crystalline MOFs or polymers), support nature (free-standing or supported), and the initial glass-former composition. All of these variations are employed to design the membrane exhibiting the molecular-sieving characteristics that specifically target one or another separation – while our membrane in particular works best for the challenging methane separations. In Figure S 5, we present a qualitative overview of the various reported MOF-glass membranes tested for different gas separations. [...]

2. “Large area” should be quantified and compared to the current state of the art.

We thank the referee for this comment and agree that this description needs clarification. Indeed, more mature polycrystalline MOF membranes can be commonly fabricated within this and sometimes larger size range. In this work, the “large-area” remark refers, first of all, to the free-standing neat MOF glass layers which have been complicated to fabricate even at the demonstrated here centimeter-scale. Second, this remark further refers to the scaling possibilities that we envision for this type of membranes particularly – e.g., modular designs in Figure 6. We have now clarified it throughout the text, for instance in the following sections:

Despite the great promise, obtaining neat and free-standing MOF glass membranes of a larger area remains challenging. [...] As a result, achieving at least typical membrane area of the polycrystalline MOF membranes – which is by itself commonly limited to the centimeter scale – while preserving porosity and therefore performance characteristics in MOF glasses remains highly problematic. Moreover, in the absence of such established procedures further scaling considerations – potentially unlocking industrially-relevant meter-scale membrane areas – remain unexplored.

[...]

Specifically, first we were able to produce large-area centimeter-scale, free-standing MOF glass membrane [...] As an outlook we will provide strategies for the further development of MOF-glass membranes towards real-world application, including a perspective approach for their upscaling to the industrially relevant membrane areas.

3. Permeation and selectivities should be quantified and compared to the current state of the art.

This data is now provided in Table S3 and Figure S5 and discussed accordingly in the main text, as described in detail in our response to the Comment 1 of this referee.

3. Is there a rationale behind the selection and order of permeability test displayed in Figure 4c,d?

Well, the permeation tests were done mainly for applications in carbon capture, H₂ purification and He upgrading. We tried to collect data that is comparable with literature and is relevant for potential applications likewise.

4. Why the possible binary gas mixtures were restricted to four (i.e. H₂/CO₂, CO₂/N₂, He/CO₂, H₂/CH₄)? Including He/CH₄ and CO₂/CH₄ separation could enhance the impact of the manuscript.

We thank the referee and add our permeation tests for He/CH₄ additionally. That was a measurement we tested long and rigorously. We stopped measuring gases against CH₄, as there was almost no difference to the single gas data. CH₄ is simply not permeating through the MOF-glass membranes undetectable, and the measurement is not giving us any more information as this that we get from single gas permeation, which we thought is not rendering the mixed gas quantification very significant.

5. Line 310ff: In the context of the mentioned “perovskite-based membranes”, it would be helpful to direct the reader by citing two relevant reviews.

We thank the referee for noting this – indeed, the relevant reference has been accidentally removed when revising the manuscript prior to submission. We added one reference that we thought is significant for the advancement of the field of perovskite membranes from recent times.

6. Supporting material, section 1.7: Avoid duplicating content in the second paragraph and use a consistent acronym for X-ray spectroscopy (e.g. EDS or EDX).

We thank the referee for this comment – the acronyms are now consistent within the document.

Reviewer #5:

This work reports a type of ZIF-62 glass membranes, fabricated through thermal expansion coefficient matching between the MOF glass and the processing devices. The authors observed impressive selectivity particularly for bulkier gas molecules, and also discussed the possible pathway forward towards the practical applications. Overall, this contribution address some of the critical challenges associated with the MOF glass, particularly ZIF-62, and the fabricated membrane could offer some solutions to challenging separations. The paper can be considered for publication in Nature Communications, after some further improvement.

The authors thank the referee for their time and positive evaluation of our work. Below, we provide a point-by-point response to all comments.

1. The authors claims the preservation of mechanically robust, cracking free membrane is originated from the CTE matching between MOF glass and Al foil. This could be the reason but more investigation will be required to fully support this claim. For example, does the ZIF-62 attach strongly with the foil after sintering? Or the mechanism could be just a low energy interface that does not provide additional stress to MOFs? More types of foils/substrates with different CTE would be welcomed.

We thank the referee for this valuable comment. First, we have now performed experiments with two additional substrates – gold and borosilicate glass (BOROFLOAT®, Schott AG). Both brittle borosilicate glass with very low CTE ($3.25 \times 10^{-6} \text{ K}^{-1}$) and ductile gold with relatively low CTE ($14.2 \times 10^{-6} \text{ K}^{-1}$) yielded similar to soda-lime glass and platinum results – ZIF-62 glasses cracked significantly because of the CTE mismatch. We would like to highlight again that although brittle material with high CTE could also potentially yield a crack-free a_g ZIF-62 and such experiment would indeed be interesting, there is no readily available materials with such characteristics that are at the same time inert and temperature-stable enough to withstand the experiment.

Figure S1 and S2 has been now added:

Figure S 1 Results of ZIF-62 melting while pressed between borosilicate glass BOROFLOAT®.

Figure S 2 Results of ZIF-62 melting while pressed between pure gold foil (0.2 mm thickness).

As well as the corresponding text:

Testing other inert substrates such as borosilicate glass BOROFLLOAT® (brittle, very low CTE of $3.25 \times 10^{-6} \text{ K}^{-1}$) and gold (ductile, relatively low CTE of $14.2 \times 10^{-6} \text{ K}^{-1}$) also resulted in significant cracking (**Figure S 1** and **S2**).

This discussion is now included in the Supplementary Note 1 and referred to in the main text.

Supplementary Note 1

Estimation of substrate-dependent residual thermal stresses in ZIF-62 glass

A rough estimate of these resulting thermal stresses can be derived on the basis of the thin film's elastic properties (elastic modulus E , and Poisson's ratio ν), and the mismatch in thermal expansion coefficients between the substrate and the film (α_s and α_f , respectively), and by the temperature difference ΔT (e.g., from the synthesis temperature down to room temperature), following:

$$\sigma_{\text{th}} = \frac{E}{1-\nu} (\alpha_s - \alpha_f) \Delta T \quad (1)$$

Such an approach can be applied to the present case of free-standing MOF glass processing to achieve semi-quantitative insights into their post-synthesis cracking behavior. First of all, it is worthy of mention that the elastic properties of ZIF-62 glasses can vary by over 25% (e.g., $6.58 \pm 0.02 \text{ GPa}$ and $5.2 \pm 0.9 \text{ GPa}$)^{5,6} from one report to another (stemming from differences in sample purity or resulting porosity after melting)⁷. Nevertheless, such preliminary values are sufficient to estimate the order of magnitude of the residual thermal stresses; hence, elastic properties of $E = 5.2 \text{ GPa}$ and $\nu = 0.343$ will be employed from ref. 1.⁶ Second, the temperature difference ΔT is the difference between the glass transition temperature T_g of $322 \text{ }^\circ\text{C}$ (at which the glass ceases to flow over the laboratory time scale) and room temperature. As a means of evaluation, the strength of ZIF-62 glasses – measured by three-point bending tests using parallelepipedic specimens, and reported to be 8 MPa by To *et al.*⁶ – is employed. Note that, to the authors' knowledge, this is the only experimentally obtained strength reported for ZIF-62 glasses and that, like their processing-dependent optical and mechanical properties, this value can be greatly affected by the employed synthesis and characterization procedures (and, notably, by the surface state following

polishing). From this, approximations of the residual thermal stresses can be estimated for each of the initially considered substrates, namely soda-lime silica glass, aluminum, and platinum and newly added borosilicate glass and gold:

Table S 1 Substrate-dependent residual thermal stresses.

Substrate	Coefficient of thermal expansion α , $10^{-6} \text{ }^\circ\text{C}^{-1}$	$\alpha_s - \alpha_f$, $10^{-6} \text{ }^\circ\text{C}^{-1}$	σ_{th}
Soda-lime silica glass	9.2	-22.9	54.7 MPa
Aluminum	24.0	-8.1	19.4 MPa
Platinum	9.0	-23.1	55.2 MPa
BOROFLOAT®	3.25	-28.8	68.8 MPa
Gold	14.2	-17.9	42.8 MPa

The obtained values of residual thermal stresses thus range in the order of a few tens of MPa, with that obtained using aluminum foil being substantially smaller than those obtained using other substrates. These values therefore compare quantitatively to the previously reported ZIF-62 glass strength, and thus support ascribing the observed cracking behavior primarily to residual thermal stresses; while the σ_{th} when using aluminum foil is close to the latter, the σ_{th} when using other substrates greatly exceeds it, resulting in the cracking of the specimens.

2. It is impressive to see the high selectivity of this membrane, which address one of the critical challenges in separation. There are some recent argument on the use of W-K setup for membrane permeation test - for example, the sweep gas of He can back diffuse to the feed side, and this will lead to a higher measured membrane permeability than the actual value. Could the author provide further discussion on this?

The referee is completely right about that. The Wicke-Kallenbach tech was originally developed for larger hydrocarbons and their diffusion on active coal, where no back diffusion occurs. In our case we are working with a porous system, in which backdiffusion can occur. We use the sweep gas, because it is very easy to bring the analyte gas (the permeate in our case) to the GC system. There are other ways in membrane science, such as pressure decay measurements, which also come with their own problems, for instance long waiting times and large sample area requirements, or the need for a mass spectrometer. In the research context, we always making assumptions. Since we are measuring with 1 ml/min sweep gas and 100 ml/min feed gas (in mixed gas 50 mln gas 1, 50 mln gas 2), the back diffusion probability of gas is very low due to a low chemical potential gradient. The sweep gas is not detectable in the retentate at these rates and we assume that back diffusion is very low and can be presumably be ignored.

3. It would be interesting to see if the membrane performance variance with different thickness, and/or different benzimidazolate ratio.

As demonstrated in other studies on MOF glass gas separation membranes (e.g., 10.1038/s41467-025-56295-x), reducing the thickness leads in the first approach to an increase in flux, but often sacrifices selectivity. This tradeoff is intuitive and is anticipated for our membrane as well; yet, the need to maintain the mechanical integrity of the free-standing glass membrane would at this stage not allow for the significant decrease in the membrane's thickness (compare 10.1002/anie.202401817). While significant changes would indeed be expected when decreasing the membrane thickness down to at least the tens-of-micrometers range, such thicknesses cannot yet be

achieved for neat free-standing MOF glass membranes by either our or other existing methods and are therefore outside the scope of this study. On a similar note, we agree with the referee that investigating influence of different linker ratios on the separation performance would be interesting and could represent another tool to adjust the performance for specific targeted separations. Yet we respectfully note that investigating this influence in a comprehensive manner would deserve a separate study.

4. There are multiple typos in the manuscript, better to revise them.

We have now carefully proof-read the manuscript and corrected the typos.

Rebuttal Letter

We thank all five excellent reviewers for the time dedicated to reviewing our manuscript and for their general evaluation of our work. We further want to thank the editor for his work and guidance.

The reviewers' comments are answered point-by-point below. The reviewers' comments are formatted in *italic*, our responses in green, changes in the manuscript in yellow, and parts removed from the main text – in ~~striketrough red~~.

The main revision are:

- We did everything requested from the editorial office, which was mainly a shortening of the abstract and following the guidelines provided throughout the AIP list, source data etc.
- We added an author contribution list.
- We have asked for a quantitative device limit from the device manufacturer and added the values to our paper in response to the comments of reviewer 3.

We are looking forward to publication of our manuscript and thank everyone involved again for their constructive feedback.

On behalf of all authors,

Dr. Alexander Knebel

Reviewer #1:

The revision and the response from the authors have addressed the questions from the referee. It can be recommended for publication in this current version.

We thank the referee for the time and effort spent on the improving our manuscript.

Reviewer #2:

The authors have well addressed the reviewers' comments. I think the revised manuscript can be accepted by Nature Communn.

We thank the reviewer for increasing the quality of our manuscript through his valuable comments, and the time committed.

Reviewer #3:

This is an excellent paper that provides a practical and scientifically rigorous solution to a long-standing problem in MOF-glass membrane fabrication. The authors successfully demonstrate the production of large-area, crack-free, self-supported MOF-glass membranes with outstanding molecular sieving capabilities. The methodology is well-detailed, and the results are compelling. Especially I like the detailed analysis and clear presentation in this paper. I have only the two comments for improving this strong paper:

We thank the reviewer for his positive evaluation of our paper.

The authors states that methane permeability was so low that it was "undetectable" by their gas chromatography setup. Therefore, please include a quantitative upper bound for the methane permeability, rather than simply stating it's "undetectable".

This section was already added to the paper during the last round of revisions; however, the detection limits and quantification limits have been asked from Shimadzu (producer of the GC device) and they stated that a quantification is usually possible until 10 ppm, while the detection limit is much lower. We have corrected our values to even lower numbers than in our previous version:

The minimum concentration of a gas in a permeate sample that can be detected by the gas chromatography system with thermal conductivity detector is 0,00009 %, while it is safe to quantify concentrations at 0.001 % (>10 ppm) or above.

The gas permeation experiments were conducted at room temperature (22°C) and 1 atm. A brief discussion or ideally, some data on how the separation performance (permeability and selectivity) changes with variations in temperature and pressure would provide a more complete (impactful) picture of the membrane's versatility and applicability in different industrial scenarios.

The referee has a good point here. We will certainly do that in our upcoming studies. The data we generated until now on p- and T-variation is too preliminary to be published with this work. The proof-of-concept for a working, large and free-standing MOF-glass membrane has been demonstrated in this paper.

Reviewer #4:

The authors addressed my concerns in a reasonable manner. The manuscript is recommended for publication.

We thank the reviewer for his valuable time and effort and that he increased the quality of our paper.

Reviewer #5:

The authors have addressed the comments raised by the last round of review - now the paper can be accepted

We thank the reviewer for the time spent on the evaluation of our manuscript and acknowledge the value added.